# Soft X-ray Fluorescence and Near-Edge Absorption Microscopy for Investigating Metabolic Features in Biological Systems: A Review

**DOI:** 10.3390/ijms24043220

**Published:** 2023-02-06

**Authors:** Valentina Bonanni, Alessandra Gianoncelli

**Affiliations:** Elettra—Sincrotrone Trieste, Strada Statale 14—km 163,5 in AREA Science Park, Basovizza, 34149 Trieste, Italy

**Keywords:** X-ray microscopy, soft X-ray, X-ray fluorescence, X-ray absorption near-edge spectroscopy, soft X-ray absorption microscopy, metabolism

## Abstract

Scanning transmission X-ray microscopy (STXM) provides the imaging of biological specimens allowing the parallel collection of localized spectroscopic information by X-ray fluorescence (XRF) and/or X-ray Absorption Near Edge Spectroscopy (XANES). The complex metabolic mechanisms which can take place in biological systems can be explored by these techniques by tracing even small quantities of the chemical elements involved in the metabolic pathways. Here, we present a review of the most recent publications in the synchrotrons’ scenario where soft X-ray spectro-microscopy has been employed in life science as well as in environmental research.

## 1. Introduction

The possibility of investigating metabolic processes impacts a wide range of research fields, from life science and medicine [1,2] to environmental and agriculture research [3,4]. Among the numerous applied techniques, spatially-resolved micro X-ray fluorescence (μ-XRF) and/or X-ray absorption near-edge spectroscopy (μ-XANES) have shown to be particularly effective in following the metabolic changes at the elemental and subcellular level.

For instance, the alterations in cellular metabolic pathways, which are evident in cancer cells compared to healthy cells, may cause deficiency or imbalance of a number of chemical elements, including low atomic weight elements, such as Na, Mg, and Al, as well as transition metals such as Fe, Cu, Zn. Depending on the elements, the mechanisms and physiological activities are different; for example, altered intracellular K and Na levels may lead to dysregulation in glycolysis [5], while deficiency of Mg, one of the principal components of the major metabolic processes, is characteristic of several types of cancer [6]. Appropriate operation of the gluconeogenesis phase that occurs in mitochondria is impacted by the proper level of copper and iron, whose changes may be associated with aggressive cancer progression [7].

Transition metals, in particular, have essential roles in brain structure and function, and several studies have confirmed the association between aberrant protein synthesis and folding in neurodegenerative disorders and disrupted metabolism of metals [8]. Iron and copper are essential for all organisms that undertake oxidative metabolism and, along with zinc, represent the three most abundant trace metals in the human brain. The ability of iron and copper to readily change oxidation state is a crucial property required for their biological functionality but, at the same time, represents a risk for their potential toxicity [9,10].

Iron and its homeostasis are intimately related to inflammatory responses in cells. During infections, available free iron is decreased and becomes sequestered in cells of the reticuloendothelial system due to inflammatory signaling cascades. The interactions between iron status and immune function are well-described but still poorly understood, particularly with regard to the molecular mechanisms that regulate these interactions [11].

In addition, the entry of calcium, through voltage-dependent ion channels, into the neurons and exchanges among intracellular components are of critical importance for brain functionality [12].

Concerning cardiovascular diseases, sodium has been shown to be strongly related to cardiomyocyte contraction, magnesium plays a fundamental role in cardiac metabolism, and iron may be involved in the induction of oxidative stress [13].

Related to this, the role of Mg is particularly evident in the functionality of cardiac, vascular and uterine muscles. Namely, elevated extracellular Mg concentration intensifies the excretion of Ca from contractile parts of muscle cells, inhibiting important enzymes that take part in (de)phosphorylation processes [14]. A direct correlation between total Mg content and distribution in tissue and colorectal cancer has been widely investigated due to the increased rate of metabolism of cancer cells and related ATP consumption [15].

Microcalcifications (MCs) are an example of a broader class of pathological mineral deposits, in which mineral formation occurs in tissues that normally do not mineralize. The link between these MCs and cancer metabolism and therapy response in breast cancers is still unknown and under investigation [16].

Sulfur, making a constitutive part of methionine and cysteine, is an essential element for all living organisms. Besides the proteogenic function, cysteine is a precursor of many biologically important molecules, such as the powerful oxidant glutathione and coenzyme A, an integral part of lipid and glucose metabolism. In pathology, it is considered essential in prostate cancer progression where, in particular, oxidative stress leads to continuous damage of critical cellular constituents, such as proteins, nucleotides, lipids, and metabolites [17].

Inorganic arsenic-related species are able to be transformed into many metabolic products, including monomethylarsonous or monomethylarsonic acid and dimethylarsonous or dimethylarsinic in hepatic stellate cells through the oxidoreduction and methylation action [18].

Besides the wide scenario of metabolism in health and diseases in humans and animals, bacterial and microbial metabolism is one of the oldest topics in microbiology [19], and it often relates also to modifications of absorbed elements, impacting several research fields such as environment and agriculture [20,21,22] up to archeology and geology [23,24].

Magnetotactic bacteria (MTB) are phylogenetically and metabolically diverse. Their metabolic versatility, diverse intracellular inclusions, and abundance within aquatic oxic-anoxic interfaces indicate a contribution of MTB to the biogeochemical cycling of Fe, O, S, C, N, and P across this important redox interface [25].

Cable bacteria are multicellular microorganisms in the Desulfobulbaceae family that display a unique electrogenic metabolism, in which electrical currents are channeled along a chain of more than 10,000 cells [26].

Biomineralization of Ca-carbonate and magnetite [27], as well as of iron disulfide (pyrite) [28,29], represent a couple of examples of the wide research field of bacteria synthesis of intracellular submicrometer-sized inorganic precipitates.

Some species of bacteria are noted to be highly tolerant to high levels of As. In particular, Fe(II)-oxidizing bacteria have recently been identified as a possible factor in As immobilization [30].

Some fungal biomineralization processes can effectively degrade organic pollutants, opening the possibility of applications of fungi in land bioremediation [31].

A broad research field of energy metabolism in fungi is focused on changes in phosphorus metabolism and distributions, spread over the several cellular compartments, and its response to the action of different kinds of stressors, where particularly vanadate as phosphorus analog takes a central position in inhibition and simulation of enzymes of glucose and glycogen metabolism. By using synchrotron-based μ-XRF imaging and XANES spectroscopy, detoxification mechanisms have been explained [32].

Phosphorus, in particular, represents a key element in fertilization, but its resource availability is declining, and new strategies for soil treatments are increasingly required [33].

Besides phosphorus, two essential components of the primary and secondary metabolism of plants, iron and sulfur, are important micronutrients required for their optimal growth and development [34]. Spatial μ-XRF investigation in combination with μ-XANES can help in understanding nutrient dynamics and bioavailability at the interface between roots of plants and soil.

Currently, the metabolic processes which take place in the soil can completely change pathways depending on soil chemical and mineralogical properties and, thus, depending on the application of organic or inorganic amendment [35].

The chemical transformation of elements and the impact of these changes have been shown to be important factors in agriculture. An example is represented by rice, one of the most consumed staple foods in the world, which is also the prime source of dietary ingestion of heavy metals. Among them, Cd and As have been shown to alter the metabolic pathways of other chemical elements, such as N or organic compounds, e.g., thiols. The localization and distribution of these heavy elements within plant tissues remain fundamental in order to investigate the risks to human health [36,37].

Ca, Cl, K, P, and S are the major chemical elements distributed in almost all tree rings, and besides the role played in metabolism, their distribution and speciation can give information on contaminants and pollutants [38].

The present review aims at highlighting the potentialities of μ-XRF and μ-XANES in the soft X-ray regime, illustrating selected scientific outcomes and also providing an overview of the worldwide available instruments and of the general scientific trend.

## 2. STXM, μ-XRF and μ-XANES

In scanning transmission X-ray microscopy (STXM), monochromatic X-rays are focused on the sample by using an X-rays lens such as a zone plate. An absorption image is generated by raster scanning the sample in the focal plane of the zone plate whilst measuring transmitted X-ray intensity for each point. When the transmitted X-ray photons are collected by a two-dimensional detector, simultaneous differential phase contrast images can be generated, complementing the absorption ones [39,40,41]. An important feature of X-ray microscopy is the possibility to integrate spectroscopic analyses at multiple length scales, till sub-micron or even nanometer range.

XRF is a characteristic X-ray emission that results from the de-excitation of the atoms ionized by the absorption of X-ray radiation. Since the de-excitation process involves discrete electron levels in the atoms, XRF spectroscopy is an excellent tool for elemental and chemical analysis. Furthermore, a low energy XRF (LEXRF) setup operated in the soft X-ray regime (50–4000 eV) is especially suited for bio-related research as it gives insight into the elemental distribution of low-Z elements (Figure 1): carbon, nitrogen, oxygen, fluorine, sodium, magnesium, phosphor and other elements, such as iron and zinc, widely involved in the metabolism of biological systems at cellular or subcellular levels [42].

Figure 1 shows the fluorescence yield as a function of the energy up to 4000 eV for the K-edges and the L-edges. These yields represent the probability of a core hole in the K or L shells being filled by a radiative process in competition with nonradiative processes. It is worth mentioning that some elements can be detected exciting different electron energy levels, e.g., iron can be probed by incident energy above its K-edge as well as above its L-edges. The more efficient edges are the K ones which are related to the most internal electron energy levels but the lower energy edges, i.e., L and M, allow the detection of heavy elements for which the K-edges cannot be accessible in the soft X-ray regime.

Synchrotron X-rays can also be tuned at energies with required precision to excite electrons from core atomic shells, causing a sensitive variation in photo-absorption at energies equal to the shell binding energy. This increase identifies an ionization edge in the X-ray absorption spectrum, and the analysis of the characteristic shape of the absorption spectrum near the ionization edge is known as XANES. Again, as for μ-XRF, the combination of μ-XANES with STXM provides localized information across the scanned area on the metal oxidation state and is sensitive to the types and numbers of coordinated ligand atoms [44]. XANES spectra can be collected in transmission or fluorescence mode, though the fluorescence mode is usually preferred for low (approximately ppm) concentrations [8].

Figure 2 shows how an XRF map of a specific chemical element and/or a stack of images taken at different incident energy (XANES) can be collected in parallel to the scanning process of an STXM measurement.

Depending on the specimen and the scientific analysis which has to be performed, one technique may be preferable to the other one. XANES has the great advantage of resolving the spectrum with a higher energy resolution, allowing the discrimination of the oxidation states of a specific element. However, the possibility of expanding the spectral range in order to probe the speciation of several chemical elements is limited by the necessity of resetting the instrument if the energy range is particularly wide. For STXM it usually implies a continued refocusing due to the chromaticity of the zone plate optics. XRF allows a simultaneous multi-elemental analysis with a tremendous sensitivity for trace elements compared to XANES, despite the loss of information on the chemical bonds.

Several beamlines dedicated to STXM and/or μ-XRF and μ-XANES techniques operate in each synchrotron facility around the world. The soft X-rays (50–4000 eV) beamlines mostly involved in research on metabolism are summarized in Table 1. They are basically able to investigate, at the micrometric or sub-micrometric length scale, the distribution and/or speciation of light elements up to the transition metals.

It is worth mentioning that there are some beamlines operating spectroscopic imaging without collecting the X-ray absorption image in transmission. The contrast image obtained by the elements map of the exposed specimens’ area allows the reconstruction of a sort of morphology of the sample. Moreover, some endstations are equipped with visible light microscopes which permit the localization of the X-rays scanned areas.

## 3. Experimental Studies

In this section, we highlight some of the recent works on metabolism performed at the synchrotron beamlines where the STXM technique has been combined with μ-XRF and μ-XANES spectroscopies. We focused not only on the wide range of applications of such techniques but also on their possible combination with complementary investigation tools and their versatility, showing both standard and less usual configurations.

In Figure 3 we have reported some bibliometric indications of the fast-growing interest over the last years in publishing works related to metabolism investigated by soft X-ray microspectroscopy. The research field of these papers’ results is heterogeneous, involving environmental science and biomedicine.

### 3.1. Life Science and Medicine

Recent studies on amyloid plaque cores isolated from Alzheimer’s disease subjects have allowed revealing the presence of metals in their elemental form in the brain. In particular, in the work of Everett J. and coauthors [10], the synchrotron-based STXM technique was used to locate amyloid plaques and establish their morphology. Then, X-ray absorption spectromicroscopy was used to determine the precise chemical state of iron and copper. The discovery of metals in their elemental form in the brain raises new questions regarding their generation and their role in neurochemistry, neurobiology, and the etiology of neurodegenerative disease. STXM experiments were performed at the Advanced Light Source (Lawrence Berkeley National Laboratory, CA, USA) 11.0.2 beamline [45] and at the Diamond Light Source (Oxfordshire, UK) I08 beamline. In these experiments, the STXM technique was employed both as imaging and spectroscopy, exploiting the different contrast which can be obtained by scanning the sample at two different energies: at the iron and copper L_3_-edges and at off-peak energy. The off-peak image was then subtracted from the peak image, yielding a Fe or Cu metals map. These two energies combination in the STXM imaging were also applied in order to distinguish oxidation states of a given metal, by exploiting the preferential absorption of X-rays in different energy ranges by specific metal phases. Finally, a more common method of the μ-XANES technique was employed collecting a series of images (called a “stack”) acquired spanning over a desired energy absorption edge. Using this approach, specific x-ray absorption spectra can be extracted from each pixel of the image, allowing the chemical composition analysis of the specimen with a highly localized (<50 nm in this instance) spatial resolution.

The role of iron is also the subject of the studies presented by L. Pascolo and coauthors [46,47] where the occurrence of asbestos fibers and ferruginous bodies in human lung tissues are investigated and correlated to inflammation processes. The asbestos bodies are not inert structures, as they cause a continuous mobilization of iron from the surrounding tissues. By tracking iron presence, together with other trace elements such as magnesium, it was possible to reveal the presence of rare uncoated fibers in the tissue which variably attracted iron from the surrounding. The identification and localization of Fe and Mg were performed by applying soft and hard XRF at three different synchrotron beamlines (TwinMic beamline at Elettra-Sincrotrone Trieste (Trieste, Italy) [48], ID21 beamline at the European Synchrotron Radiation Facility (ESRF, Grenoble, France) [49] and XFM beamline at the Australian Synchrotron [50]). The study was completed with high spatial resolution STXM at TwinMic and with μ-XANES on Fe K-edge at ID21, paving the way for multi-beamlines complementary analyses.

The observation of living cells under biocompatible wet conditions represents an interesting in-situ approach to observing real-time changes in various biochemical processes in an environment close to physiological conditions. In Ref. [51], the authors presented a new STXM system suitable for this kind of experiment. The setup employs a liquid-enclosing graphene (LGS) system, which provides a biocompatible environment over a few hours to live cells by perfectly confining the liquid without leakage. Taking advantage of this LGS- STXM system, the authors observed the chemical variation of oxygen species from a single wet cell and analyzed absorption spectra near the oxygen K-edge (530–560 eV). The experiment was performed at the 10A beamline at the Pohang Light Source [52]. In this case, the μ-XANES analysis was employed collecting stacks of STXM images by changing the incident photon energy while keeping the focal position at the sample plane. A XANES spectrum can then be extracted from a specific point or area of the image. Similarly, C. Arble and coauthors [53] have developed and tested a graphene encapsulation liquid cell (GrELCs) designed to be suitable for in situ studies of biological samples with an array of analytical techniques such as optical/fluorescence/infrared as well as scanning electron microscopy and X-ray microscopy. The performances and robustness of the cell were evaluated by Fourier-transform infrared spectromicroscopy and by STXM coupled with LEXRF, demonstrating that GrELCs are able to contain very thin water layers for several hours even in high vacuum environments, with the water layer standing above the cells, thanks to the conformability of the graphene layers.

X-ray micro-spectroscopy can be helpful also in detecting the topical penetration of drugs within the skin. Through the relationship between light penetration and the drug concentration, it is possible to determine the amount of drug in different skin layers. In particular, K. Yamamoto and coauthors have applied X-ray microscopy to follow the penetration of topically applied tacrolimus formulated in micelles in inflamed murine skin [54]. The experiment was performed at the Swiss Light Source at the PolLux instrument [55]. The element selectivity of the X-rays allowed probing separately of the distribution and penetration depth of drugs and carrier molecules into biological tissues. Images were recorded at selected photon energies below and at the C K-edge (270–320 eV) as well as below and at the O K-edge (500 eV and 550 eV). From preliminary absorption spectra of the species under study, it was possible to individuate two energies at which there was a change in differential absorption primarily for one component, but not for fixed skin. A similar investigation method was employed at the UVSOR III synchrotron radiation facility (Institute for Molecular Science, Okazaki, Japan) using the BL4U beamline [56]. This label-free spectromicroscopy has thus proved to be a valuable tool to probe drug-loaded carriers and their penetration in human skin [57].

### 3.2. Microbiology and Bacteriology

Sulfur, together with its ions, represents one of the main elements involved in microbiology studies. Here we present two different works involving microorganisms in sulfur compounds.

The role of sulfur- and sulfate-reducing bacteria (SRB) in the kinetics of pyrite formation and in the regulation of Fe, S, and P biogeochemical cycles is the topic of the study of Ref. [29]. Pyrite is the main sedimentary sink for sulfur over geological time scales but the role of microorganisms in pyrite formation is still lacking. The chemical characterization of the evolution of the minerals was monitored employing several complementary techniques. In particular, STXM at the Fe L_2,3_-edges and C K-edge was performed at the beamline HERMES at SOLEIL (Saint Aubin, France) [58]. Image map stacks were recorded at 288.2 and 280 eV for C and image stacks were then collected from 690 to 740 eV at the Fe L_2,3_-edges. The results of this study seem to suggest that pyritization is indeed regulated by SRB, enhancing phosphate release into the aqueous phase by increased efficiency of iron sulfide precipitation.

In Ref. [24], the authors have shown that iron sulfide biominerals precipitated in the presence of sulfate-reducing microorganisms incorporate and adsorb organic molecules, leading to the formation of stable organo-mineral aggregates that could persist for years in anoxic environments. STXM and NEXAFS spectroscopy were used to analyze the distribution of Fe, S, C, O and N and the chemical speciation of Fe, C, O and N in solid phases. Measurements were carried out at the 11.0.2.2 beamline of the Advanced Light Source (Berkeley, CA, USA) [45] and at the SM beamline of the Canadian Light Source (Saskatoon, SK, Canada) [59]. The stacks of images acquired across the absorption edges of the elements of interest were converted from transmission scale to optical density scale, by subtracting images acquired at energies before the absorption edges. Spectra recorded across the C K-edge were fitted following a specific procedure described in the work, which allows quantifying the contribution of peaks associated with individual functional groups. Moreover, correlations between the distribution of Fe and C in mineral aggregates were evaluated by applying a scatter plot method.

As shown in Figure 4, the research fields involving metabolic pathways are not always strictly separated and for some works, it can be difficult to allocate them in a specific macroarea. Some microbiological processes, in particular, can be deeply related to agriculture and environmental science, e.g., concerning microbial degradation of organic matter and soil nutrient transformations or biotransformation in pollutants and toxic substances.

### 3.3. Environment and Agriculture

An example of a work that combines microbial activity with climatic and environmental changes impacts is that of V. Fichtner et al. [60]. Aragonite minerals extracted from *Arctica islandica* mosculls and skeletal hard parts of *Porites* sp. were used to examine the impact of microbial metabolic activity on sulfur content that may have a significant bias in the estimation of global sulfur cycling. Synchrotron analyses were performed at the PHOENIX beamline at the Paul Scherrer Institute (PSI, Villigen, Switzerland), where chemical imaging with micrometric spatial resolution can be performed by employing both XANES and XRF techniques. In particular, this study was carried out not applying the two spectroscopic techniques in parallel to a scanning imaging measurement. The localization of the spectroscopic signal is obtained by collecting fluorescence and transmitted light images and focusing the X-ray beam to a spot size of a few μm. The data clearly identified a distinct sensitivity of organically bound sulfur in biogenic aragonite to microbial alteration, and X-ray microanalysis allows localizing the sulfur in specific areas of the bivals and corals.

A great variety of sulfur species and their specific function in a number of molecules in environmental and biological systems requires the usage of advanced techniques capable of identifying and distinguishing them in a more accurate way. In Ref. [61], the authors have presented a coupled μ-XRF and XANES spectroscopy method for determining the presence of specific sulfur species in coral tissues and skeletons at high spatial resolution. The method, already introduced by Mayhew et al. [62], is based on synchrotron-radiation μ-XRF maps collected at multiple energies (ME) within the specific element edge. By conducting principal components analysis (PCA) on the ME maps, the μ-XANES signal can be optimized and performed just in specific spots. Through an iterative process that involves XRF and XANES fitting, the authors were able to identify and discern several reduced sulfur species, such as glutathione disulfide, cysteine, and sulfoxide, as well as organic sulfate as represented by chondroitin sulfate. The experiment was performed at the 14-3 beamline at the Stanford Synchrotron Radiation Lightsource (SSRL).

A more standard concurrent application of μ-XRF and XANES spectroscopies has been presented in the paper of Yan B. and coauthors [63], where XRF chemical mapping was conducted at the two different energies (3570 eV and 7200 eV) in order to avoid effects of potassium interference with Cd L lines. Successively, Cadmium L_3_-edge μ-XANES spectra were collected in the areas of interest. The experiment was performed on the ID21 scanning X-ray microscope at the European Synchrotron Radiation Facility (ESRF, Grenoble, France) [49]. The main objective of this work was the characterization of the distribution of Cd and nutrients (notably Cu, Fe, Mn, P, S and Zn) in the durum wheat grain. The combination of μ-XRF and μ-XANES allowed exploring the presence of Cd with sulfur ligands, suggesting that sulfur fertilization could be a way to reduce the allocation of Cd to the grain as well as its accumulation in the endosperm in durum wheat.

A similar employment of the STXM and μ-XRF combination was applied in the study of P. Zhang et al. [64] performed at the BL08U1 beamline of the Shanghai Synchrotron Radiation Facility (SSRF) with the intent to investigate biotransformation and translocation processes of Ce nanoparticles in plant systems. First, a dual-energy method was performed on the selected regions of the sample to obtain the spatial distribution of cerium components. Then, image sequences (stack) were acquired at multiple energies spanning the relevant element absorption edge (from 884 to 915 eV for Ce M_4,5_ edge). The study seems to indicate a particle dissolution promoted by organic acids.

Biotransformation and biochemical modifications, in general, have a strong impact on the environment, reflected in pollution, toxification and detoxification processes. In the study of D. Medas et al. [65], zinc incorporation into marine bivalve shells grown in mine-polluted seabed sediments was investigated with the intent to make a step closer to understanding bivalve biomineralization and its significance for environmental monitoring and paleo reconstruction. Fe and Zn distributions were studied at the TwinMic beamline at Elettra-Sincrotrone Trieste (Trieste, Italy) [48], whereas Ca and S distributions were investigated at the I08-SXM beamline in Diamond Light Source (Didcot, UK). Different biogeochemical mechanisms involved in the regulation of specific elements’ content and their chemical speciation can be unraveled by the simultaneous application of XRF and other complementary techniques. In particular, this study indicates that bivalves have developed different biogeochemical mechanisms to regulate Zn content and its chemical speciation and that these two parameters are indeed interconnected.

N. Yamaguchi et al. have recently raised attention to a complementary analysis combining chemical extraction and spectroscopy in the soil phosphorus speciation study, especially in relationship with the implementation of animal manure compost in agriculture [35]. In this study, speciation of P in soil microsites was established after applying traditionally used fertilizer and a mixture of original and nitrogen-deprived fertilizer with cattle compost. These data were compared with average P speciation in the corresponding bulk soil. The XRF microscale analysis was performed at the Tender Energy Spectroscopy (TES) beamline (8-BM), NSLS-II, Brookhaven National Laboratory (Upton, NY, USA) [66] and at BL27SU, SPring-8 (Japan) [67]. The elemental maps of P, S, and Si were obtained by the on-the-fly scan mode with the detection of fluorescence yield by a Canberra Ultra-low-energy Ge detector. This XRF modality differs from the usual point-by-point scan, given that the sample is moved continuously, with data being collected while the sample is in motion. Fly scan is a much faster strategy, but it may be affected by some inevitable motion blurring of the detail in the image. Together with the XRF analysis, Phosphorus K-edge μ-XANES on selected P hot spots were obtained by the partial fluorescence yield method.

## 4. Conclusions

STXM, combined with spectroscopic μ-XRF and μ-XANES, offers the unique advantage of simultaneously acquiring imaging and elemental components/structural information with a sub-micrometric spatial resolution. With a view to studying cellular metabolisms, where chemical elements are at the center of various chemical reactions, the applications of these techniques represent a powerful method to explore the fate and role of various compounds at the subcellular level.

Although STXM spatial resolution can be several orders of magnitude inferior to Transmission Electron Microscopy (TEM), the chemical sensitivity is generally higher than the TEM- Electron Energy Loss Spectroscopy (EELS). In particular, the sensitivity of soft X-ray STXM to biology-related elements, such as C, N, O, S, and Ca. Moreover, the sample preparation requirements for STXM analyses are less demanding than for TEM ones.

Nevertheless, similarly to all techniques, STXM cannot be considered exhaustive, as it can provide specifically defined information, which should be complemented as much as possible by other complementary approaches. As for other analytical methods, the accessible chemical elements are intrinsically limited as they are determined by the specific energy range. Moreover, in the soft X-ray regime, restrictions do exist in terms of sample size, sample thickness and acquisition time, even if there is a global trend to overcome these limitations both improving the experimental environment [51,53], the light sources [68] and the detectors [69,70,71,72,73,74], and employing some expedients to reduce the sampling frequency or to automatize the acquisition process [75,76].

The potentiality to study in vitro or in vivo cellular systems remains the main goal for investigating biological processes, especially in the soft X-ray regime. Beyond the relatively low tolerance towards ionizing radiation of living organisms, other aspects, such as working in a liquid environment together with vacuum conditions, make this objective hard to be reached. However, several efforts have been made in the past, especially towards the designing and the development of in-situ methods and the extent to in situ–in vivo measurements seems potentially possible.

With this overview of the active microspectroscopy synchrotron-based laboratories and their involvement in the vast published literature during the recent period, the authors intend to provide some indication of the experiments that can be performed and which kind of information can be obtained from them in many research fields that involve elemental metabolic pathways.

## Figures and Tables

**Figure 1 ijms-24-03220-f001:**
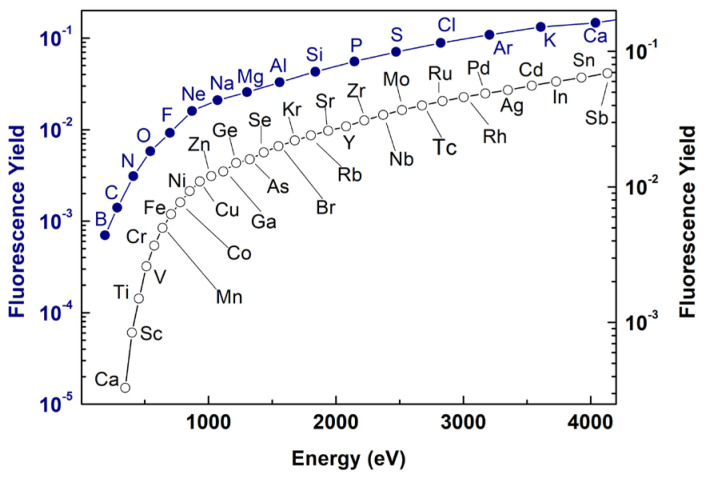
Elements accessible by the soft XRF technique. The blue curve (full circles) shows the fluorescence yield (left axis) as a function of the energy for the K-edges. The black curve (empty circles) shows the fluorescence yield (right axis) for the L3-edges. The data are tabulated on the basis of the database reported in [43].

**Figure 2 ijms-24-03220-f002:**
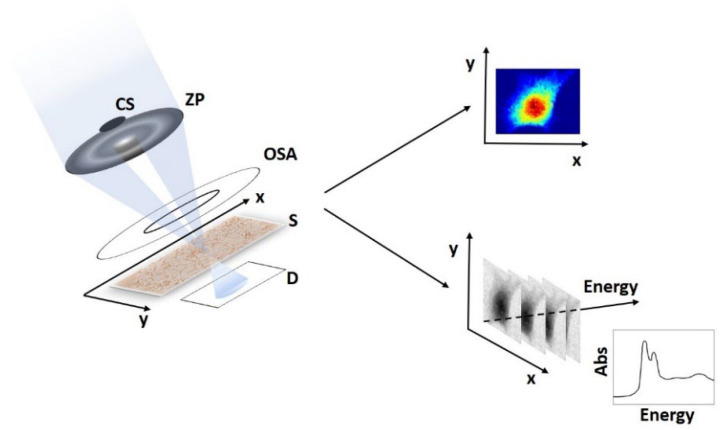
A sketch of combining STXM (**left**) with the two spectroscopic techniques: XRF (**top-right**) and μ-XANES (**bottom-right**). CS indicates the central stop; ZP the zone plate; OSA the diffraction order-selecting aperture; S the specimen and D the detector.

**Figure 3 ijms-24-03220-f003:**
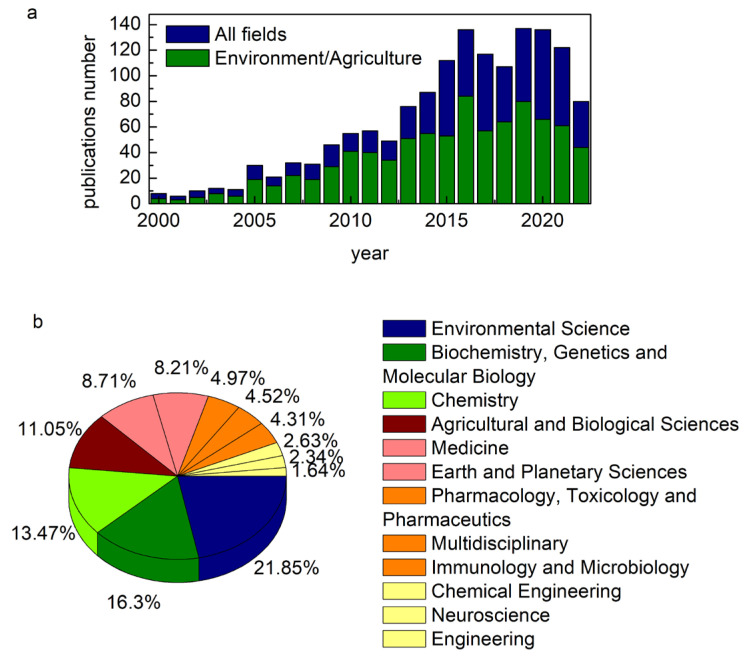
The number of (**a**) papers published between 2000 and 2022 related to metabolism pathways investigated by μ-XRF and μ-XANES spectroscopies in all the research fields (blue) and in Environmental and Agriculture Science (green) and (**b**) distribution of the same papers among Scopus classification (Source Scopus).

**Figure 4 ijms-24-03220-f004:**
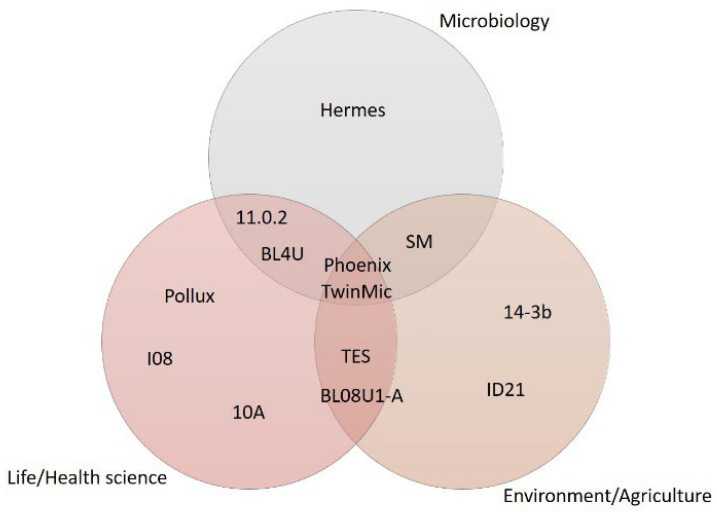
A scheme of the synchrotron beamlines where it is possible to perform soft X-ray micro-spectroscopy in relationship with the three main research fields involving metabolic pathways.

**Table 1 ijms-24-03220-t001:** List of synchrotron soft X-ray beamlines suitable for research on metabolism. In the table are indicated some specifics for each beamline. The application field refers to previous publications correlated to metabolism.

Beamline	Facility	Country	Available Techniques	Spot Size/Spatial Resolution ^#^ [μm]	Energy Range [eV]	Application
11.0.2	Advanced Light Source (ALS)	USA	STXM, μ-XANES	0.025	160–2000	Life/Health science, Microbiology
MYSTIIC @EMIL *	Bessy II	Germany	STXM	0.025^#^	250–1500	
SM	Canadian Light Source (CLS)	Canada	STXM, μ-XRF and μ-XANES	0.03	130–2700	Microbiology, Environment/Agriculture
I08	Diamond	UK	STXM, μ-XRF and μ-XANES	0.02^#^	250–4200	Life/Health science
TwinMic	Elettra	Italy	STXM, μ-XRF and μ-XANES	0.1	400–2200	Environment/Agriculture, Microbiology, Life/Health science
ID21	European Synchrotron Radiation Facility (ESRF)	France	STXM, μ-XRF and μ-XANES	0.07 × 0.03	2100–9200	Environment/Agriculture
SoftiMAX *	MAX IV	Sweden	STXM, μ-XRF and μ-XANES	0.06	275–2500	
TES	National Synchrotron Light Source II (NSLS II)	USA	μ-XRF and μ-XANES	2	2000–5500	Environment/Agriculture, Life/Health science
Phoenix	Paul Scherrer Institute (PSI)	Switzerland	μ-XRF and μ-XANES	2.5	400–2000	Environment/Agriculture, Microbiology, Life/Health science
Pollux	Paul Scherrer Institute (PSI)	Switzerland	STXM	0.02^#^	250–1600	Life/Health science
10A	Pohang Light Source (PLS)	Korea	STXM, μ-XANES	0.025^#^	100–2000	Life/Health science
BL08U1-A	Shanghai Synchrotron Radiation Facility (SSRF)	China	STXM, μ-XANES	0.03^#^	250–2000	Life/Health science, Environment/Agriculture
Demeter *	Solaris	Poland	STXM		10–2000	
Hermes	Soleil	France	STXM, μ-XANES	0.025	70–2500	Microbiology
14-3b	Stanford Synchrotron Radiation Lightsource (SSRL)	USA	μ-XRF and μ-XANES	5	2100–5000	Environment/Agriculture
27A *	Taiwan Photon Source (TPS)	Taiwan		0.03^#^	90–2500	
BL4U	UVSOR	Japan	STXM	0.03	50–770	Microbiology, Life/Health science

* these beamlines have started to operate recently or are under design or construction.

## Data Availability

Not applicable.

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
