# Peer review of "Soft X-ray Fluorescence and Near-Edge Absorption Microscopy for Investigating Metabolic Features in Biological Systems: A Review"

_ijms, 2023, doi:10.3390/ijms24043220_

Round 1
Reviewer 1 Report
In the manuscript entitled "Soft X-ray fluorescence and near-edge absorption microscopy for investigating metabolic features in biological systems: a review", the authors gave an overview of the active microspectroscopy synchrotron-based laboratories and their involvement in the vast published literature during the recent period, providing some indication on the experiments that can be performed and which kind of information can be obtained.
The manuscript is well organized and the description of the published literature is presented in a very good way.
In my opinion, the manuscript should be accepted for publication after minor revision, in particular:
- Figure 1. A definition of the Fluorescence yield, as well as how it can be measured and calculated should be added (directly in the footnote or in the main text).
- Page 4, line 142. When the "low-energy XRF" and "soft-X rays" are introduced, it should be added an interval of characteristic energies and/or wavelengths (frequencies).
- Figure 3: The addition of a third graph, highlighting the variation of publications in the various areas (or macroareas) could be interesting, to show if the growth in publication is related to an increase of interest in a particular area of research.
- The discussion of the publications should be slightly changed, because it is presented as a list of results. I suggest to describe better for each work presented the aim and the conclusions, to add some figures to have a better readable manuscript and to try to connect the different parts of the text.
- The conclusions in my opinion lack of the presentation of the drawbacks of the discussed techniques, as well as a better discussion of the impact that this approach could have and a discussion of future perspectives.
Reviewer 2 Report
The Review article on "Soft X-ray fluorescence and near-edge absorption microscopy for investigating metabolic features in biological systems: a review" by Bonanni and Gianoncelli is well written represents important information on the progression of analytical technology in metabolic analysis. However, the review is two focused on one specific method. It will be better to expand the review to a family of methods. The others show show perspective and progress in rersearch that lead to the development the latest versions of XANES and XRF systems. I recommend extensive revision to this effect. The paper should be downscaled to a normal paper on the "Soft X-ray fluorescence and near-edge absorption microscopy for investigating metabolic features in biological systems", NOT a review. Diagrams and figures from other sources should be properly referenced and proof should be made available to show that permission is granted to reproduce in IJMS.
Round 2
Reviewer 1 Report
The Reviewer want to thank the authors for their kind answers to the comments.
In the present form the manuscript should be accepted for publication.
Reviewer 2 Report
I noticed very minor errors that I believed will be corrected before the author submits his thesis.
Regards